# Double-pentagon silicon chains in a quasi-1D Si/Ag(001) surface alloy

Conor Hogan [1,2] ✉, Andrea Sette[2], Vasil A. Saroka [2], Stefano Colonna [1], Roberto Flammini [1], Laurita Florean[3,4], Romain Bernard[3], Laurence Masson[5], Geoffroy Prévot[3] & Fabio Ronci [1] ✉

Silicon surface alloys and silicide nanolayers are highly important as contact materials in integrated circuit devices. Here we demonstrate that the sub-monolayer Si/Ag(001) surface reconstruction, reported to exhibit interesting topological properties, comprises a quasi-one-dimensional Si-Ag surface alloy based on chains of planar double-pentagon Si moieties. This geometry is determined using a combination of density functional theory calculations, scanning tunnelling microscopy, and grazing incidence x-ray diffraction simulations, and yields an electronic structure in excellent agreement with photoemission measurements. This work provides further evidence of pentagonal geometries in 2D materials and heterostructures and elucidates the importance of surface alloying in stabilizing their formation.

A surface alloy is a two-dimensional (2D) mixture of a metal with other elements that is confined within a few atomic layers on a surface, being immiscible in the bulk phase[1–3]. Surface alloys are of particular relevance in heteroepitaxial growth and in heterogeneous and single site catalysis[4–6] and commonly form well-ordered 2D layers[7–9] and nanostructured phases[10–14]. Of recent interest is their functionality as a precursor or even a scaffold for 2D growth of $MoS_2$[15] and monoelemental Xenes like tellurene[16], stanene[17,18], thallene[19], antimonene[20], as well as silicene, the silicon analogue of graphene[21–24]. Along with nanoscale silicides, surface alloys of silicon are of particular technological relevance due to their use as a contact material in microelectronic devices[25].

Surface alloy formation typically begins with the ejection of substrate (metal) atoms from the top layer followed by insertion and/or substitution of adsorbate atoms[3]. The silicon-silver interface is a prototypical system for studying this atom-exchange mechanism. During the growth of Si/Ag(111), the topmost layer of Ag is replaced by a continuous sheet of silicene[26–29]. In contrast, on Si/Ag(110)[27,30], the initial ejection of Ag atoms accompanies the formation of pentagonal Si clusters[31], seeding the growth of a surface alloy of pentagonal Si nanoribbons in Ag missing rows[32–34], before leading at higher coverage to the appearance of

a dumbbell silicene layer[35], predicted to constitute the true ground state of 2D silicon[36].

The phenomenon and relevance of surface alloying in the Si/Ag(001) system has drawn recent specific attention[37]. The first experimental studies were performed nearly two decades ago by Leandri and coworkers[38–40]. They reported that Si deposition results in the formation of a low coverage (3 × 3) phase based on 1D tetramer stripes with subsequent formation at higher Si coverage of a complex striped superstructure having a hexagonal pattern. Although the precise atomic structure of the latter remains unknown[41,42], a simple surface alloy geometry of the (3 × 3) phase was recently proposed and identified as a potential model system for studying novel physical phenomena of confined systems, specifically the Su-Schrieffer-Heeger model of topological excitations[37].

In this work, we elucidate the atomic structure and electronic properties of the (3 × 3) reconstruction of Si/Ag(001) using a combination of particle swarm optimization, atomistic thermodynamics within density functional theory (DFT), scanning tunnelling microscopy (STM) measurements, and DFT simulations of surface X-ray diffraction (SXRD) and angle-resolved photoelectron spectroscopy (ARPES) measurements. Like for the other two silver surfaces, silicon deposition is shown to cause silver atom ejection from the topmost

[1]CNR-Istituto di Struttura della Materia (CNR-ISM), Rome, Italy. [2]Dipartimento di Fisica, Università di Roma "Tor Vergata", Rome, Italy. [3]Institut des NanoSciences de Paris, Sorbonne Université, CNRS-UMR 7588, Paris, France. [4]Laboratoire de Chimie Physique Matière et Rayonnement, UMR7614, Sorbonne Université, CNRS, Paris, France. [5]Aix Marseille Univ, CNRS, CINaM, AMUtech, Marseille, France. ✉e-mail: conor.hogan@cnr.it; fabio.ronci@cnr.it

layer of Ag(001), leading to a Si/Ag surface alloy with a notable quasi-1D character. The new geometry is built from the same pentagonal motifs identified during growth of Si/Ag(110)[31], emphasising the ubiquity of pentagon geometries in surface alloys and in precursors to growth of 2D materials.

## Results

### STM and LEED study

The (3 × 3) phase is obtained by Si deposition on the clean Ag(001) substrate held at about 490 K, as clearly demonstrated by the LEED pattern in Supplementary Fig. 1, recorded after 0.8 ML Si deposition (see Methods for definition of ML). The corresponding STM images, reported in Fig. 1a, b, confirm the formation of (3 × 3) domains separated by evident bright linear features. In addition, brighter four-lobe features are randomly observed in the domains, most probably induced by the presence of defects, in agreement with the literature[40]. The image in Fig. 1b shows several interesting features. The single (3 × 3) domain is characterised by a regular square grid of unit cells, each containing 4 bright spots. As highlighted by the cyan grid, adjacent (3 × 3) domains are mostly observed to be in phase each other, i.e., their unit cell grids are overlapping. Furthermore, adjacent domains are rotated by 90° to each other, as clearly visible in the magnified images of the two areas delimited by dashed white squares, in which the (3 × 3) unit cells are indicated by the green squares and the yellow arrows highlight the presence of vertical (horizontal) dark elongated

troughs in the right (left) domain. Hence, the symmetry of the (3 × 3) domains is not $D_4$, but rather $D_2$. This is also confirmed by the high-resolution STM image reported in Fig. 1c, showing that the four lobes inside the square unit cell are arranged in a 0.44 × 0.48 nm$^2$ rectangular pattern. (These values are different from those reported in ref. 40 (0.41 × 0.28 nm); we ascribe this to inferior resolution in the earlier work).

The presence of flat Ag(001) terraces in the upper left corner of Fig. 1a and in the left side of Fig. 1b indicates a partial local Si coverage. This is also confirmed by the STM images reported in Supplementary Fig. 2a, c in which a larger flat terrace is observed. Interestingly, bright (dark) round atomic-size features are observed on the flat Ag(001) terrace in the filled (empty) state images, recalling the single atom Si substitutional defects already observed in Si/Ag(111)[26]. Most importantly, such Ag(001) terraces have a greater apparent height than the adjacent (3 × 3) domains, as shown by the line profiles in Supplementary Fig. 2b, d, suggesting that the (3 × 3) domains grow inside the topmost substrate atomic layer rather than on top of it, causing the ejection of Ag atoms. This evidence is not surprising, as it was already observed for Si deposition on Ag(111)[27,28] and Ag(110)[27,30].

In order to consolidate the hypothesis that Ag atom ejection upon Si deposition occurs in Si/Ag(001) as well, we performed real-time STM measurements, showing the evolution of the same area of the Ag(001) surface upon Si deposition. The two images in Fig. 1d show that Si deposition initially (left panel) results in the formation of square-

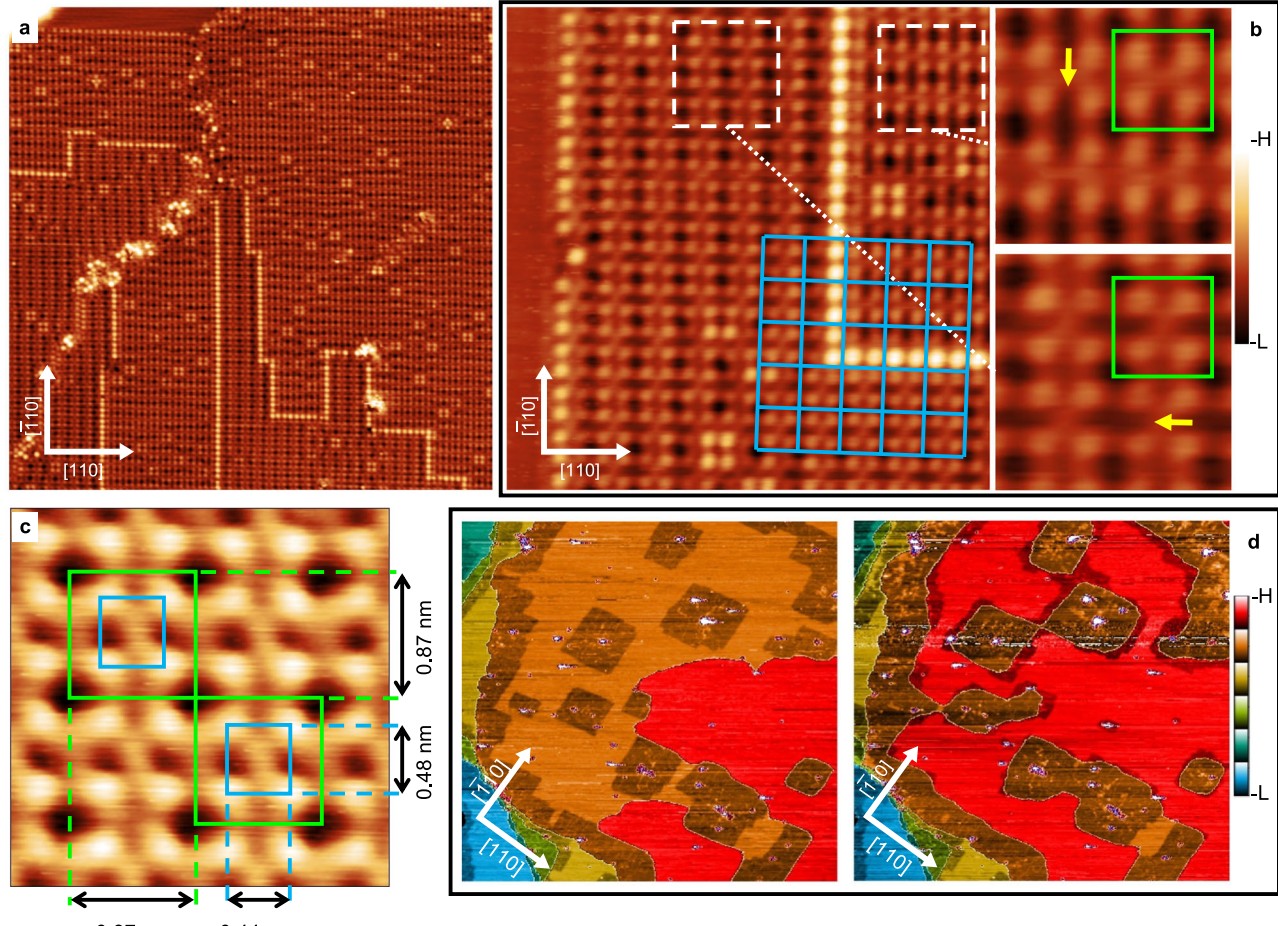

**Fig. 1 | STM images of the 0.8 ML Si/Ag(001) surface collected at 80 K.** **a** 40 × 40 nm$^2$ STM image ($V_s$ = −1 V, $I$ = 2 nA). **b** 10 × 10 nm$^2$ STM image showing two adjacent (3 × 3) domains ($V_s$ = −1 V, $I$ = 2 nA). The superimposed cyan grid shows the common lattice of the two domains, whose unit cells are highlighted with green squares in the two blowups. **c** High-resolution 2.6 × 2.6 nm$^2$ STM image ($V_s$ = 0.01 V, $I$ = 20 nA) highlighting the $D_2$ symmetry of a (3 × 3) domain. **d** Real-time STM images showing the evolution, from left to right, of the Ag(001) surface during sub-monolayer Si deposition at 490 K. (287 × 276 nm$^2$, $V_s$ = 2 V, $I$ = 20 pA).

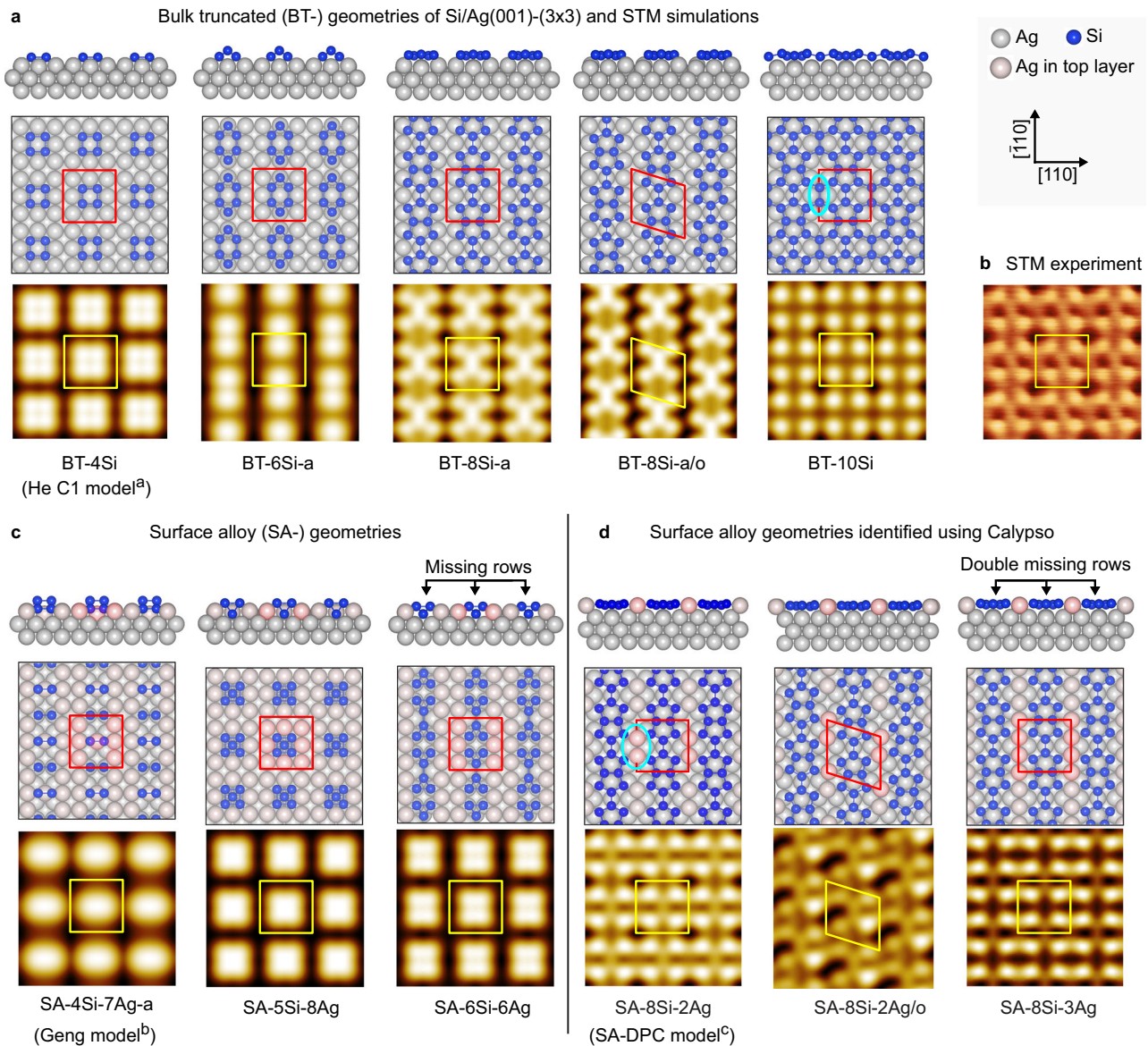

**Fig. 2 | Structural models of the Si/Ag(001)-(3 × 3) reconstruction. a** Optimized geometries of selected bulk-truncated models (top and side views) and corresponding simulated STM images (bias −1.0 eV). The (3 × 3) unit cells are indicated by red or yellow squares within a larger (9 × 9) Ag(001) unit cell area. **b** Experimental STM image at −1.0 eV bias (2.6 × 2.6 nm²). **c** Selected surface alloy geometries. **d** Surface alloy geometries identified using Calypso. Single and double missing Ag row reconstructions are present in the SA-6Si-6Ag and SA-8Si-3Ag models. Models with rhomboid unit cells (suffix '/o') do not have (3 × 3) symmetry. Ovals highlight Si dimers and Ag pairs. [a]From He, ref. 43; [b]From Geng et al., ref. 37; [c]This work. SA-DPC: surface alloy-double pentamer chain.

shaped recessed islands in the Ag(001) terrace and of stripes at terrace edges, all having lower apparent height with respect to the Ag(001) plane. Simultaneously, Ag outgrowths (depicted in red) are formed on the Ag(001) terraces. As Si coverage proceeds (Fig. 1d, right panel), the recessed islands increase in size and coalesce, while a clear expansion of the upper terrace (in red) is observed, confirming Ag atom ejection.

**Geometry, formation energy and STM simulations**

From the experimental results discussed thus far, we establish several criteria that must be taken into account when constructing and evaluating plausible (3 × 3) models: (i) full coverage is obtained above 0.8 ML, i.e., the (3 × 3) unit cell must contain more than 7 Si atoms; (ii) in spite of the square unit cell, (3 × 3) domains are characterised by a $D_2$ symmetry; (iii) the model must comply with the observed rectangular 4-lobe pattern in experimental STM images; (iv) the (3 × 3) unit cells of

a single domain are always observed to be in phase with each other, forming a regular square grid, indicating that a relevant coupling between them must exist in both orthogonal directions; (v) given the demonstrated Ag atom ejection upon Si deposition, possible reconstruction of the substrate should be considered. Optimal models should obey all five criteria.

In Fig. 2, we report optimized geometries obtained via DFT calculations for selected models of the Si/Ag(001)-(3 × 3) reconstruction. Corresponding STM simulations at a bias of −1.0 eV are shown below each model. The phase diagram of relative formation energies is shown in Fig. 3a. A more complete set of investigated models is reported in Supplementary Figs. 3 and 4, with the corresponding phase diagram shown in Supplementary Fig. 5.

We first consider systematic Si adsorption on the bulk truncated surface. We adopt the nomenclature BT-$n$Si-x where $n$ indicates the number of Si atoms per unit cell (the Si coverage $\theta_{Si}$ is thus $n/9$ ML),

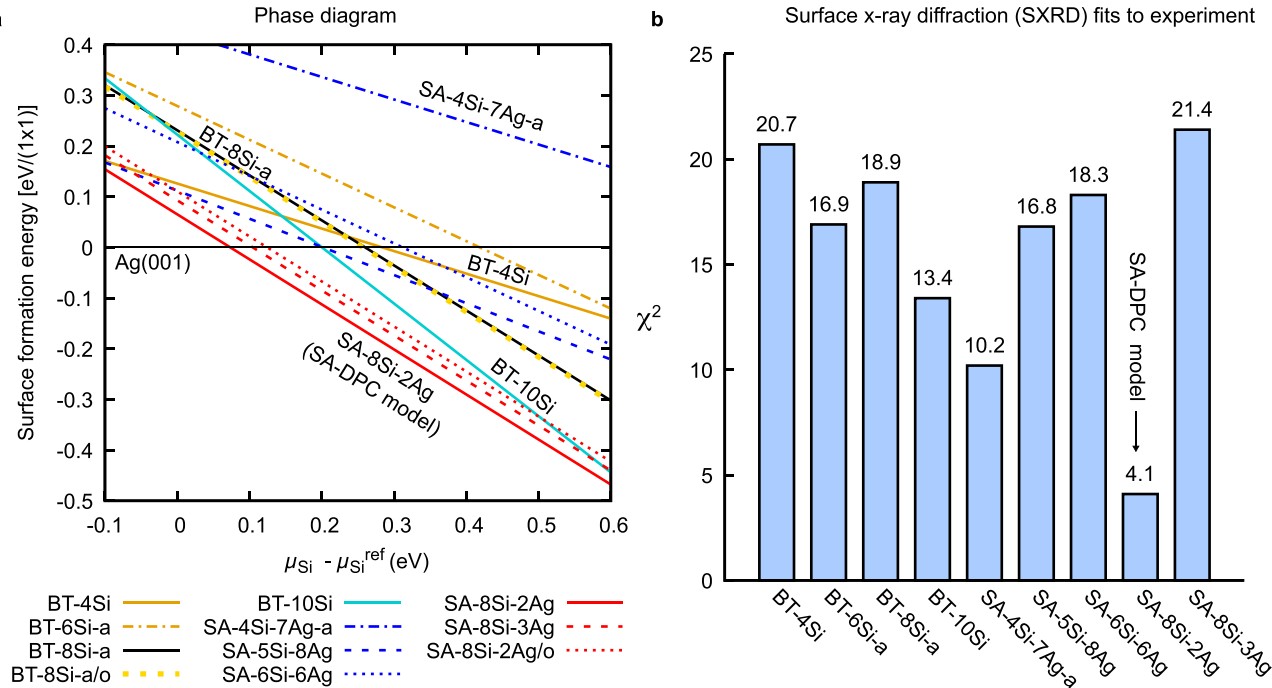

**Fig. 3 | Identifying the correct structural model. a** Surface formation energy phase diagram for the models reported in Fig. 2, relative to that of the clean Ag(001) surface. Source data are provided as a Source Data file. **b** Surface x-ray diffraction fits to the experimental data of ref. 40.

and the suffix '-x' (=a,b,c...) distinguishes different models for the same coverage. It is easy to construct a model with four protruding Si atoms (criterion (iii)), as Si tends to form dimers or small clusters. This was noted also in the work of He (BT-4Si for example was called C1 therein)[43]. However, no geometry with coverage below $\theta_{Si} = 8/9$ yields a satisfactory comparison with the experimental image shown in Fig. 2b. The BT-8Si-a model is a promising candidate, fulfilling criteria (i)–(iii). Based on two fused pentamers, it echoes the pentamer chain model of nanoribbons[32,44] and small magic clusters[31] formed on Si/Ag(110). The agreement with experiment is good: in addition to the four bright spots, a signal from the central dimer along [$\bar{1}$10] is also (weakly) discerned in the measured image. Nonetheless, it must be discarded, as it is energetically degenerate with the out-of-phase configuration (BT-8Si-a/o), thus contradicting (iv). This degeneracy would imply the presence of domains with both in-phase and out-of-phase unit cell rows, as observed for instance with silicon nanoribbons grown on Ag(110)[45]. Adding more Si atoms resolves this problem, at the cost of a slightly elevated coverage of 1.1 ML. The result is an interesting 2D adlayer network of silicon. This BT-10Si model is more stable than the others discussed previously thanks to the steeper formation energy in Fig. 3a, and its simulated STM image is also in good agreement with experiment.

All the above models neglect possible substrate reconstruction (criterion (v)). We therefore considered surface alloys (termed SA-$n$Si-$m$Ag, where $m$ is the number of Ag atoms in the topmost atomic layer), based on single or double missing rows of Ag (as found on Si/Ag(110)). Most of these models suffer the same problem as BT-8Si-a: since there is no apparent coupling between adjacent chains, we expect to observe adjacent rows also appearing out-of-phase. Moreover, the formation energies are typically higher and the simulated STM images are no better than that of BT-4Si. At this point we decided to consider more complicated geometries with variable Si/Ag stoichiometry. As a trial-and-error approach quickly becomes prohibitive, we turned to a more efficient and systematic method for exploring the surface reconstruction phase diagram and carried out a global structural search using a particle swarm optimization (PSO) algorithm as implemented in the Calypso code[46,47].

In addition to the geometries already reported above, the PSO analysis identified several potential stable or metastable models, a selection of which are indicated in red in Supplementary Figs. 3 and 4. The more promising ones, according to their symmetry and total energy, are reported in Fig. 2d and derive from the BT-8Si-a model with additional Ag atoms between the rows. For instance, the SA-8Si-3Ag model is equivalent to a BT-8Si-a model within a double missing row reconstruction. As shown in Fig. 3a, these surface alloys are much lower in energy than any other model discussed so far.

The SA-8Si-2Ag model was identified as the most stable of all considered geometries. It is similar to BT-10Si but with two Ag atoms instead of two Si atoms added to BT-8Si-a (see ovals in Fig. 2). This model contains eight Si atoms per (3 × 3) unit cell, hence full coverage of the 3 × 3 phase is obtained at 8/9 ML of Si coverage, fulfilling criterion (i). The model's symmetry is $D_2$, with Si chains running along the [$\bar{1}$10] (or the energetically equivalent [110]) substrate direction, as requested by criterion (ii). Its simulated STM image clearly shows that the rectangular 4-lobe pattern observed in experimental STM images is determined by the four Si atoms at the sides of the double pentamer chain. The position of such four atoms describes a rectangle with the expected aspect ratio (longer sides along the chain direction), satisfying criterion (iii). The symmetrical positioning of the added Ag atoms with respect to the double pentagon motif makes this geometry more stable than one with out-of-phase rows (see SA-8Si-2Ag/o), fulfilling criterion (iv). Finally, the presence of two Ag atoms in the surface layer satisfies criterion (v) as well. The SA-8Si-2Ag model thus thoroughly fulfils all the required criteria and appears thermodynamically stable. We hence propose it as the atomic structure of the Si/Ag(001)-(3 × 3) surface reconstruction. Hereon, we will refer it as the SA-DPC (surface alloy-double pentamer chain) reconstruction. In the following sections we provide definitive confirmation by direct comparison with other experimental measurements, namely bias-dependent STM, SXRD, and ARPES.

Before continuing, we remark that the few models reported in the literature do not comply with the above constraints. Leandri and coworkers[40], on the basis of STM and surface x-ray diffraction measurements, proposed a model with two tilted Si dimers (i.e., 0.44 ML

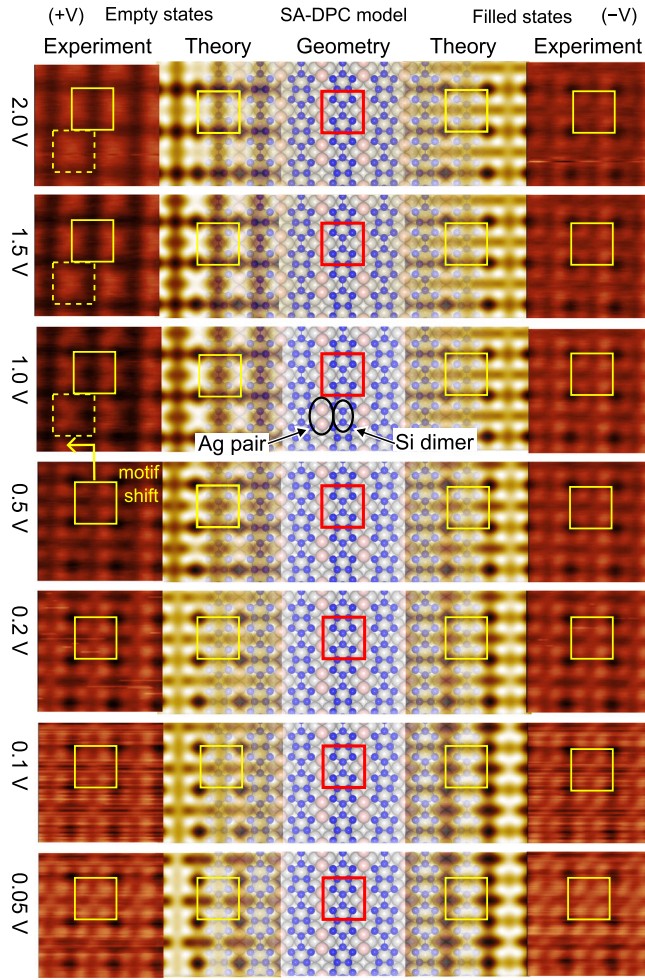

**Fig. 4 | Bias-dependent STM.** Comparison between simulated (light colour) and experimental (dark colour) 2.6 × 2.6 nm² STM images of the (3 × 3) reconstruction as a function of applied bias. Left columns: empty state (positive sample bias) images. Right columns: filled state (negative sample bias) images. Experimental images collected at 80 K with tunnelling current I = 20 nA. Solid squares highlight the (3 × 3) unit cell. The SA-DPC model is shown in the centre column and partially overlays the computed images.

coverage) on an unreconstructed Ag(001) substrate. Our DFT simulations show that such dimer pairs spontaneously relax to form flat tetramers as in the metastable BT-4Si model. Guo·min He[43] theoretically investigated a large number of geometries for the (3 × 3) phase. None of the models satisfactorily explain the STM images, however. Furthermore, a c(4 × 4) cell was considered, which is in conflict with LEED and STM measurements. Very recently Geng and coworkers[37] proposed a two-dimer Si model with $D_2$ symmetry atop a reconstructed Ag substrate, on the basis of DFT calculations and simulations of ARPES measurements. Their model is reported in Fig. 2c as SA-4Si-7Ag-a. It contradicts criteria (i) and (iv), and its formation energy is very high (see Fig. 3a). Indeed, we found that the model is geometrically stable only if symmetry constraints are imposed. Allowing the geometry to relax freely yields the clearly incorrect SA-4Si-7Ag-b model (see Supplementary Fig. 4b) containing a four-atom Si cluster.

## Bias-dependent STM

To ensure a robust validation of the SA-DPC model, we performed high-resolution, bias-dependent STM measurements. Indeed, the selection of specific bias values may often result in a fortuitous match between experimental and simulated images even for unreliable

models. On the other hand, a good match in a wide bias range provides a strong validation to the model, as some of us demonstrated in ref. 48.

Figure 4 reports a series of bias-dependent 2.6 × 2.6 nm² experimental STM images collected at 80 K (outer columns), along with the corresponding series of simulated STM images obtained from the SA-DPC model (inner columns). A partially overlaid ball-and-stick model of the SA-DPC model is reported in the central column as a reference. The experimental images in Fig. 4 were cropped from the larger 7.5 × 7.5 nm² images reported in Supplementary Fig. 7, in which a typical domain boundary between two adjacent 90° rotated (3 × 3) domains is displayed.

The simulated images match extremely well to the experimental ones. Inspection of the simulated filled-state images reveals that the four lobes correspond to the four Si atoms at the sides of the double pentamer chain. The low bias images appear similar both for filled and empty states (compare the three lowest rows), and allow to observe the Si dimer connecting the fused pentamers in the centre of the unit cell. Most notably, an apparent lateral shift of the four-lobe motif is observed in empty state images for bias values higher than 0.5 V. This apparent shift, made evident by the highlighted unit cell (dashed squares) and clearly observed in both the simulated and experimental images, is due to a relative local density of states (LDOS) increase on the pair of surface Ag atoms at the expense of the Si dimer (indicated in the centre of Fig. 4). This LDOS increase is confirmed by calculating the projected density of states (PDOS), reported in Supplementary Fig. 6, that shows a clear, prominent peak in the Ag surface PDOS around + 0.6 eV. This atom-specific feature in the bias-dependent STM is a strong indication of the correctness of the SA-DPC model.

## SXRD simulations

In order to test the validity of the different models with respect to the SXRD results reported by Leandri et al.[40], we have computed theoretical structure factors from the configurations obtained by DFT optimizations. For this purpose, we have used only a limited number of free parameters. As in ref. 40, two different scale factors for the fractional order and the integer order reflections have been used. The former is slightly larger to account for parts of the surface that could be unreconstructed. The only other free parameters are the sets of Debye−Waller (DW) factors for Si and Ag atoms along the in-plane and out-of-plane directions. For the sake of simplicity, Debye−Waller factors are chosen equal for Si (Ag) atoms of the same layer. For bulk atoms (i.e., for atoms below the first unreconstructed Ag layer), Debye−Waller factors have been set to a value of 0.63 Å², as in ref. 40. The agreement between experimental and simulated ($F_{th}$) structure factors is estimated by the value of

$$\chi^2 = \frac{1}{N_{pts} - N_{par}} \sum_{N_{pts}} \left( \frac{F_{th} - F_{exp}}{\sigma_{exp}} \right)^2 \qquad (1)$$

where $N_{pts}$ = 121 is the number of experimental structure factors, $N_{par}$ is the number of free parameters and $\sigma_{exp}$ is the experimental uncertainty.

Corresponding $\chi^2$ for relevant configurations are given in Fig. 3b (and Supplementary Fig. 8, for the complete set of models). The SA-DPC (SA-8Si-2Ag) model gives the best agreement with the experiments, yielding a value of $\chi^2 = 4.1$, with only 11 free parameters. For this model (see Fig. 5), theoretical structure factors along the reconstruction rods (with fractional indices) display an excellent agreement with experimental ones. A good agreement is also found for substrate rods (with integer indices) and in-plane structure factors, taking into account the limited number of free parameters. None of the other tested models correctly reproduce the rod profiles for fractional indices, even those having a moderate $\chi^2$ value.

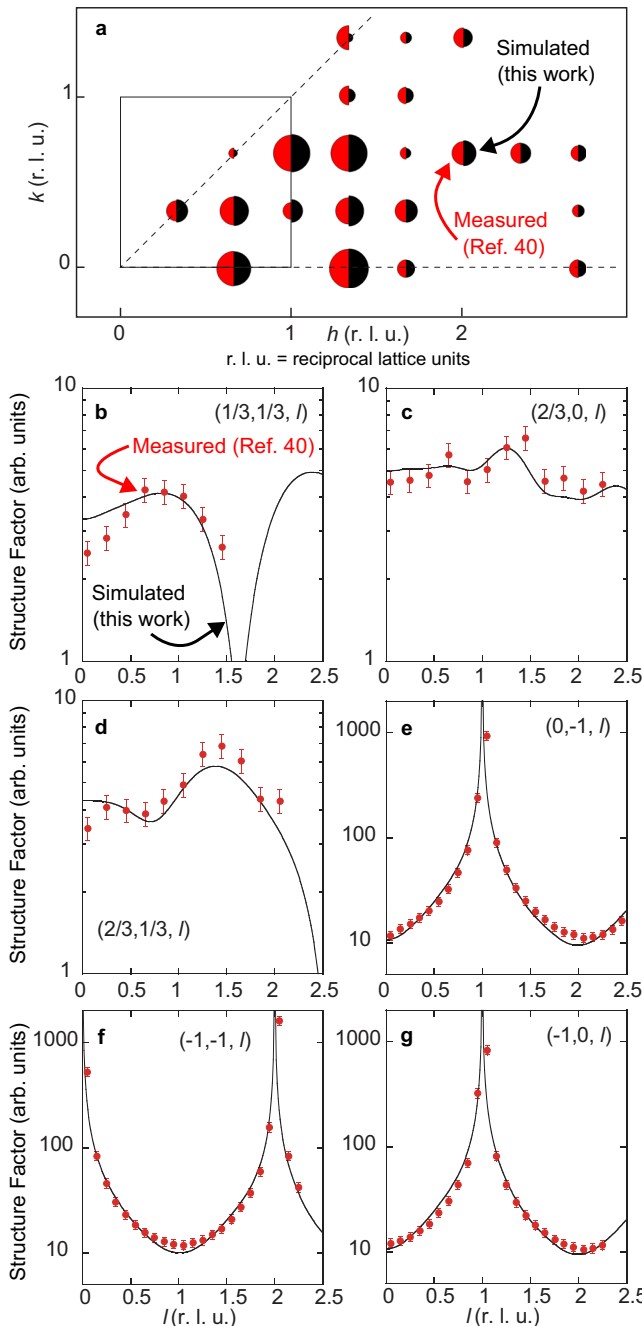

**Fig. 5 | SXRD simulations. a** Non-equivalent fractional order surface x-ray diffraction reflections of the (3 × 3) superstructure. The measured and simulated structure factor amplitudes (using the SA-DPC geometry) are shown by the area of the red and black half circles respectively. Experimental data, including standard deviation error bars, are taken from Leandri et al. (ref. 40). The conventional LEED (1 × 1) reciprocal surface cell is indicated. **b**–**g** Rod profiles for fractional and integer order reflections. The indices are given with respect to the non-reconstructed unit cell. Measured and simulated structure factor values are shown by the red dots and the black solid lines, respectively. The reciprocal lattice units (r. l. u.) for the (*h*, *k*, *l*) indices correspond to the Ag(001) surface unit cell ($a = b = 0.2889$, $c = 0.4085$). Source data are provided as a Source Data file.

## Band structure simulations

ARPES measurements of the Si/Ag(001)-(3 × 3) surface reconstruction have recently been reported by Geng et al.[37]. Selected data from that work are reproduced in Fig. 6c, e along specific cuts within the (1 × 1) Brillouin zone (see Fig. 6a) for reconstructed Si/Ag(001). Alongside the

experimental data, we report in Fig. 6d, f our band structure calculations using the SA-DPC model (Fig. 6b). The (3 × 3) bands have been unfolded along the same cuts as the ARPES data using the `unfold-x` code[49] and projected onto Si orbitals in order to allow direct comparison with the experiment.

The computed bands of the SA-DPC model are found to be in excellent agreement with the experimental data regarding both the energetic position and lineshape. We note good reproduction of the dispersive band along [$\bar{1}10$] (Cut 4) having a maximum at the $\overline{X}_{Ag}$ point, as well as the more weakly-dispersing m-shaped band along the perpendicular direction (Cut 2). The latter's energy position and bandwidth (from −1.1 eV at $\overline{X}_{Ag}$ to −0.5 eV at $\overline{M}_{Si}$) is in excellent agreement with experiment (from −1.1 eV to −0.6 eV), and clearly improves on the results of the Geng (SA-4Si-7Ag-a) model[37], which spans a range from −1.2 to −0.9 eV. Other weak features appearing in the computed bands can be attributed to remnants of bulk states of the finite Ag slab. These calculations thus constitute further independent evidence of the SA-DPC model.

## Discussion

In the previous section, we have proposed a surface alloy model of the Si/Ag(001)-(3 × 3) reconstruction. The SA-DPC model is composed of pairs of fused Si pentamers linked by single Si-Si bonds to form a continuous nanostructure on the Ag substrate, stabilized by pairs of Ag atoms within the same atomic layer. The alloy appears thermodynamically stable and yields results in excellent agreement with experimental STM, SXRD, and ARPES measurements. We also note that core level spectra[39] measured at 0.8 ML reveal three distinct Si2p components, one being twice as intense as the other two. This is consistent with the three distinct atomic environments of the eight Si atoms present in the double-pentamer motif: two Si atoms, each bound to three Si atoms, in bridge sites at the apices of the double pentamer, or in hollow sites at its centre, respectively; and four Si atoms at the double pentamer sides, each bound to two Si atoms and a surface Ag atom.

It is interesting to note the similarity between the pentamer pairs in the SA-DPC model and the magic Si clusters[31] that act as precursors for Si nanoribbon (NR) growth on Ag(110), as well as the fused pentagonal chain geometry of the Si NRs themselves[32,44]. The (3 × 3) reconstruction is thus the third example of double-pentagon Si nanostructures appearing on a Ag substrate. (No such geometry has been reported for Si on Ag(111), neither experimentally nor in theoretical studies of the initial stages of silicene formation[50], although some proposed models of 2D silicon on Al(111) contain pentagon motifs[51].) Double pentagons are thus now established as key motifs in Si nanostructures, from 0D clusters[52], 1D chains (Ref. 32 and the present work), isolated 2D sheets[36,53,54], and 3D phases (from silicon clathrates to new predicted geometries[55,56]). Although several 2D networks formed from pentagonal silicon were investigated in this work (namely BT-10Si and BT-8Si-b, see Supplementary Fig. 3), none were found to be as stable as the surface alloy SA-DPC. So far, therefore, no pure silicon *adlayer* based on pentagons has been realized: for such high Si concentrations, honeycomb silicene is apparently more stable[35]. Finally, we note that the corrugation (maximum vertical displacement difference) of the silicon atoms in the SA-DPC reconstruction is very low, about 0.22 Å, and can thus be considered a truly planar pentamer geometry. Planar pentamers of silicon have been reported as universal building blocks for the (110)-family of silicon surfaces (see ref. 57 and references therein). In contrast, the pentagonal nanostructures formed on Ag(110) are highly buckled, with corrugations of about 0.8 Å, reflecting the geometrical constraints imposed by the missing row substrate reconstruction in that case.

The planar pentamer geometry on Si/Ag(001) is stabilized by pairs of Ag atoms in the topmost atomic layer that act to lower the surface formation energy (Fig. 3a). The real-time STM measurements, Fig. 1d,

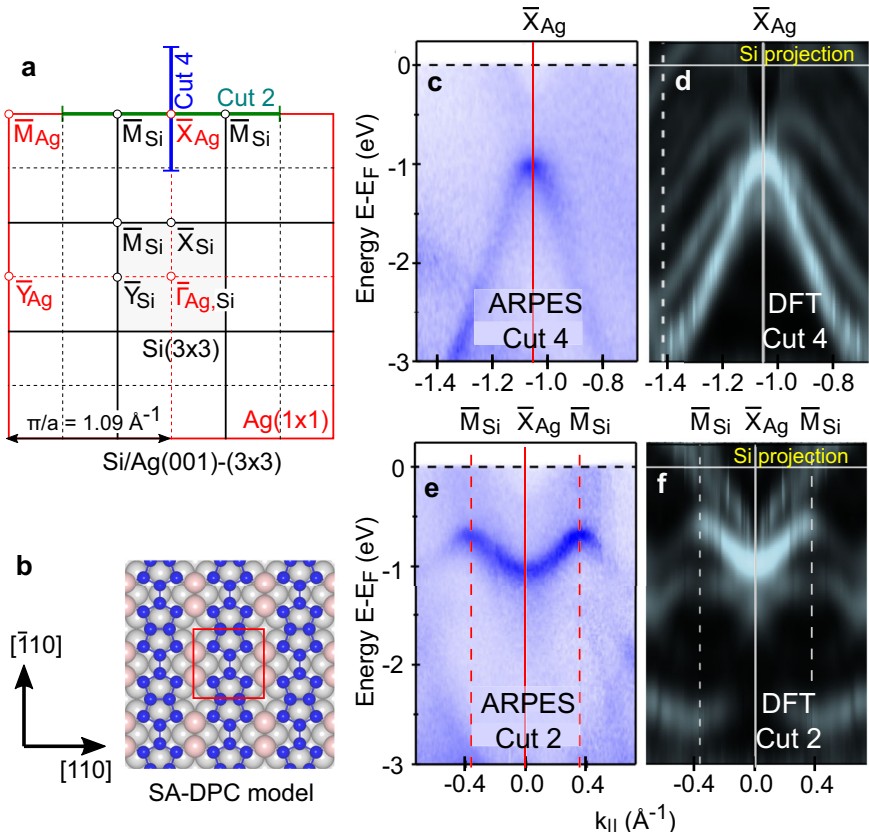

**Fig. 6 | Band structure calculations of the SA-DPC model. a** Surface Brillouin zones of Si/Ag(001)-(3 × 3) (in red) and Ag(001)-(1 × 1) (in black). **b** Structure and orientation of the SA-DPC model. **c, e** ARPES measurements on Si/Ag(001) along the cuts indicated in **a**. Reproduced (with modified annotations) from Figs. 3j, g, respectively, of Geng, D., Zhou, H., Yue, S. et al., Nat. Commun. **13**, 7000 (2022) (ref. [37]) and licensed by CC BY 4.0. **d, f** Band structure calculations of the SA-DPC model, unfolded into the (1 × 1) cell.

demonstrate that large scale etching of the topmost Ag(001) layer is needed to form the final Ag-Si surface alloy having 8(2) Si(Ag) atoms per unit cell. Similarly, the magic clusters on Si/Ag(110) form in a Ag divacancy, while nanoribbons on the same surface form on a single missing row reconstruction. As the substrate modification is confined to a single atomic layer, with Si atoms occupying non-substitutional sites, such reconstructions can be considered non-pseudomorphic Ag-Si surface alloys[2]. Surface alloys are commonly known as precursors for growth of other more stable structures. In the Si/Ag(110) case, the magic cluster Si/Ag alloy acts as a seed of nanoribbon growth; in Si/Ag(001), the (3 × 3) Si/Ag alloy forms before the reported 7 × 4 complex phase of unknown geometry[40].

Finally, we consider a pertinent question: does the SA-DPC reconstruction have a quasi-1D or a 2D character? As noted in the STM measurements and simulations, adjacent chains in the same domain always appear in-phase, suggesting some coupling between them which we attribute to the presence of the pairs of surface Ag atoms. The character of this lateral interaction can be studied further by looking at the electron localization function (a dimensionless quantity having values between 0 and 1), which offers a quantitative measure of the local electron pair density[58]. In Fig. 7, we plot the in-plane and out-of-plane ELF profiles for SA-DPC and for the related BT-8Si-a and BT-10Si models. The in-plane profiles show the pattern of Si-Si covalent bonds (ELF ~ 0.95) running throughout the double pentamer structures. In the BT-10Si case, a continuous 2D network of bonds is observed. The planar geometry suggests a predominantly $sp^2$-like character throughout. In BT-8Si-a, the blue troughs indicate decoupling of parallel chains, as previously demonstrated via the

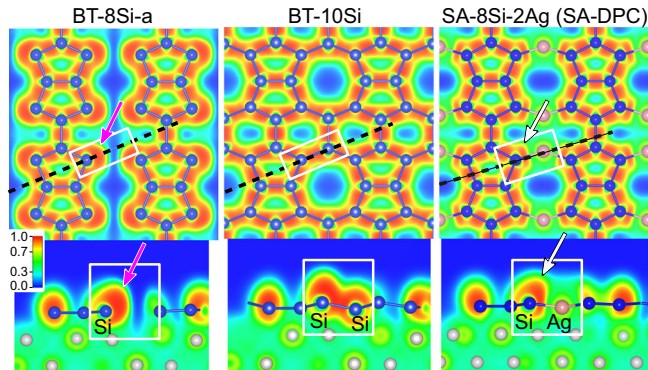

**Fig. 7 | ELF calculations.** Electron localization functions of the BT-8Si-a and related models, through a horizontal plane cutting the Si layer (top) and the vertical plane indicated by the dashed line (bottom). Arrows highlight the change in the Si lone pair charge profile when surface Ag atoms are present.

energetically degenerate BT-8Si-a/o model. These troughs are interrupted by Ag atoms in the SA-DPC model. The minimum ELF value of ~ 0.45 along the Si-Ag axis indicates a metallic bond that is nonetheless more electron-rich than a Ag-Ag bond in the bulk layers (minimum ELF ~ 0.35). The side profiles help to complete the analysis. In particular, the large lobes outside the external four Si atoms in BT-8Si-a (magenta arrows) indicate the presence of lone pairs (ELF ~ 0.95), and their out-

of-plane extension suggests $sp^3$ orbital character. When bonded to a third Si atom in BT-10Si, these atoms elevate slightly from the substrate, the lone pair shifts more towards the surface normal and the atoms become more $sp^3$-like. For SA-DPC the lone pair is notably flattened (white arrows) away from the surface Ag atoms in a pushback effect (Pauli repulsion). The near vertical lobes correspond to the four dominant spots in the measured STM images. Thus, while weak in-plane Si-Ag bonds are present to stabilize the SA-DPC geometry, the pentamer chains appear somewhat electronically isolated from their neighbours.

This character is also reflected in the computed band structures. The weakly dispersive m-shaped band with clear $(3 \times 3)$ periodicity occurs along the $\overline{M}_{Si}$-$\overline{X}_{Ag}$-$\overline{M}_{Si}$ direction (Cut 2), perpendicular to the chain orientation, and reflects localized states with a weak inter-chain coupling. Instead, the strongly dispersive band along Cut 4 runs parallel to the chain direction and suggests a more delocalized state. From an electronic point of view, therefore, the chains may be regarded as quasi-1D. We expect that the interpretation of the Si/Ag(001) surface reconstruction in terms of a 2D Su-Schrieffer-Heeger model[37] needs to be re-evaluated in light of the new SA-DPC geometry and its quasi-1D nature.

In summary, we have presented a joint experimental-theoretical study of the $(3 \times 3)$ reconstruction of the Si/Ag(001) surface. Using a combination of particle swarm optimization and DFT calculations guided by high-resolution STM measurements, we identified a new structural model that is more thermodynamically stable than any other model examined here or proposed in the literature. It yields excellent agreement with measured high-resolution STM images and in particular exhibits the correct bias-dependent behaviour, which is shown to derive from the presence of Ag atoms in the surface layer. It also yields excellent agreement with SXRD measurements, out-performing all other tested models, and with previously published ARPES spectra. The proposed model is a non-pseudomorphic Si/Ag surface alloy composed of a planar double pentagon motif linked to form quasi-1D Si chains, stabilized by the presence of the Ag atom pairs. This work reinforces the relevance of surface alloying in the epitaxial growth of 2D and quasi-1D heterostructures, and provides further evidence of the ubiquitous presence of pentagonal geometries in 2D materials and heterostructures[59,60].

## Methods

### Experiment
Si/Ag(001) samples were prepared and measured both at the Istituto di Struttura della Materia (ISM) in Rome and at the Institut des NanoSciences de Paris (INSP). Ag(001) monocrystals were cleaned in ultrahigh vacuum (UHV) conditions by repeated cycles of Ar$^+$ sputtering and annealing at 780 K. In both ISM and INSP, Si was deposited on the clean Ag(001) substrate held at (490 ± 10) K (where the error accounts for possible calibration differences in the two setups) using the same kind of Si evaporator (Focus GmbH EFM3 electron bombardment source) which allows a careful control of the Si flux. From the Si flux, carefully calibrated using a quartz crystal thickness monitor[34], silicon coverage values could be easily derived. Auger electron spectroscopy (AES), low energy electron diffraction (LEED) and STM measurements were performed to verify the quality of the Ag(001) substrate before and after silicon deposition. Coverage is reported throughout this paper in terms of monolayers (ML), defined as the Ag(001) surface atomic density, i.e., 9 atoms in a $(3 \times 3)$ unit cell.

The STM measurements performed at ISM were conducted using an Omicron LT-STM equipped with a tungsten tip and operating in a UHV chamber with a base pressure in the $10^{-10}$ mbar range at a controlled temperature of 80 K in order to minimise thermal drift issues. The STM piezo was calibrated using the Ag(001) surface as reference. We underline that all STM images reported here were only corrected for background by applying a standard plane subtraction using the Gwyddion software[61].

Real time STM imaging was performed at INSP using an Omicron VT-XA STM. The installation of the Si source in the STM chamber allows Si deposition on the STM stage and the acquisition of STM images on the very same sample area upon Si evaporation. The pressure was $1 \times 10^{-10}$ mbar during the experiments. Comparison between images of the same area performed at different times have been made by carefully correcting the STM images for drift, using a home-made procedure[62].

### Theory
Calculations of the geometry and formation energy of Si/Ag(001) surfaces were carried out using density functional theory (DFT) in the local density approximation (LDA). We use a framework of planewave/norm-conserving pseudopotentials as implemented in the quantum-ESPRESSO code[63]. The Si/Ag surface was modelled using periodically repeating supercells of asymmetric slabs, seven Ag layers thick, separated by vacuum regions 30 Å thick. The backmost three layers of Ag were fixed at the bulk positions. We use a kinetic energy cutoff of 50 Ry and the LDA lattice constant of $a_0 = 4.075$ Å. Gamma-centred $8 \times 8 \times 1$ **k**-point meshes were used with a smearing[64] of 0.05 eV. The geometry optimization threshold was 5 meV/Å. This scheme was used successfully in our previous studies of Si adsorption on other Ag surfaces[34,44,65]. For band structure calculations, we used a thicker, symmetric, 11-layer slab.

The formation energy of the Si/Ag slab is obtained from the grand canonical potential as

$$\gamma = (E_{Ag/Si} - N_{Ag}\mu_{Ag} - N_{Si}\mu_{Si})/A \tag{2}$$

Here $E_{Ag/Si}$ is the total energy of the Si/Ag(001) system, $\mu$ the chemical potential, and $A$ the unit cell area. Bulk silver is assumed to be the reservoir so we set $\mu_{Ag} = E_{Ag}^{bulk}$. In practice, we plot $\gamma^{slab}(\mu_{Si})$ in terms of a reference value, $\Delta\mu_{Si} = \mu_{Si} - \mu_{Si}^{ref}$, which we here take to be the energy of a Si atom in the bulk phase, and plot the relative formation energy of the Ag/Si reconstruction with respect to that of the clean Ag(001) slab.

Formation energies are computed for a wide range of structural models and stoichiometries, shown in Supplementary Figs. 3 and 4. Initial guesses were constructed using a mixture of experience and trial-and-error, guided by the high resolution STM images. However, this procedure did not yield a completely satisfactory geometry. To identify other possible models, we thus used the particle-swarm optimization technique for surface optimization as implemented in the Calypso code[46,47]. In a preliminary stage, we fixed the Ag slab and computed the optimal geometries for a range of Si/Ag stoichiometries using the DFTB+ code[66] with appropriate Si-Ag hopping parameters[67,68]. We then chose the best new candidate models based on criteria suggested by the STM measurements (coverage, symmetry, etc.), and fully recomputed the geometries and formation energies using quantum-ESPRESSO. Geometries obtained in this way are reported in Fig. 2d and indicated in red in Supplementary Figs. 3 and 4.

## Data availability
The data that support the findings of this study are available within the article and Supplementary Information. Source data (phase diagrams, SXRD structure factors, STM line profiles, PDOS calculations) are provided with this paper. The optimized geometries of all models generated in this study are provided in the file Supplementary Data 1. Source data are provided with this paper.

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

## Acknowledgements

C.H. acknowledges the CINECA award under the ISCRA initiative, for the availability of high-performance computing resources and support. Special thanks to Pietro Bonfá and Andrea Ferretti for help with the `unfold-x` code. S.C., R.F. and F.R. acknowledge the valuable technical support of Giovanni Emma, Antonio Olivieri and Massimiliano Rinaldi. V.A.S. was partly supported by HORIZON EUROPE Marie Sklodowska-Curie Actions (2021-PF-01, Project No. 101065500, TeraExc).

## Author contributions

C.H., L.M., G.P. and F.R. conceived the idea and designed the study. C.H. and A.S. performed the DFT calculations. STM measurements were performed by F.R., S.C. and R.F. (at CNR-ISM) and by G.P., L.F. and R.B. (at INSP). G.P. carried out the SXRD analysis. V.A.S. and R.F. analysed the electronic structure. C.H., G.P. and F.R. prepared the figures and wrote the manuscript with contributions from all authors.

## Competing interests

The authors declare no competing interests.
