## [Transparent Peer Review file · Nature Communications]

Double-pentagon silicon chains in a quasi-1D Si/Ag(001) surface alloy

Corresponding Author: Dr Conor Hogan

Version 0:

Reviewer comments:

Reviewer #1

(Remarks to the Author)

Using STM and DFT calculations the 3×3 reconstruction formed by Si adatoms on the Ag(100) is elucidated. Key point of the research is the careful analysis of the large number of possible atomic structure models, with various number density of Si as well as Ag atoms, based on the DFT calculations which enabled the authors to find, most likely, the “ground state” model. The validity of the model is confirmed based on the available SXRD Ref. 40 and ARPES Ref. 37 data. Although the pentagonal Si arrangement is not original (it is proposed as the structural element of the Si/Ag(110) reconstruction and also of the atomically clean Si(110) “ 16×2 ” reconstruction) the work remind me in some sense the work of K. Takayanagi in which the DAS model to describe the reconstruction of an atomically clean surface of Si(111) 7×7 was proposed.

Overall, I think this is a nice piece of work and would recommend its publication in Nature Communications.

The following is a list of comments:

As not all of the integer order spots are visible in the LEED pattern showed in Fig. 1S indicate the indexes of inter order spots. As only $(+1,0)$, $(0,+1)$ but not $(+1,+1)$ the pattern is unclear from the first sight.

Reviewer #2

(Remarks to the Author)

The authors present here an extensive experimental and theoretical study of the (3×3) reconstruction of the Si/Ag(001) interface. Starting from the deposition of 0.8 ML Si on Ag(001) and LEED and high resolution STM measurements, they have designed several models to determine the corresponding structure observed at the interface. As such, they have proposed numerous atomic geometries probed by DFT calculations, and formation energy determination. Also, they have refined the model considering first the possibility of Si-Ag alloy and then variable stoichiometry, through particle swarm optimization. These calculations allows the determination of a phase diagram, STM images, SXRD and ARPES simulations, in very good agreement with the experimental results. Finally, the most favorable structure appears to be a Si/Ag alloy composed of a planar double pentagon motif which forms a quasi-1D Si chain.

To my opinion, this is a very good work, with a lot of calculations in order to determine the correct structure. Comparison with literature is made all along the manuscript, and the results are very convincing owing to the numerous experimental characterizations used to corroborate theory and experiment. Therefore, I recommend this article for publication in Nature Communication without any restriction.

Reviewer #3

(Remarks to the Author)

This paper reports the geometry of submonolayer silicon chains on Ag(100) to be a 1D double-pentagon chained structure, confirmed by a combination of density functional theory calculations, scanning tunneling microscopy, and grazing incidence x-ray diffraction simulations. This work is a high-quality and timely contribution to the geometrical structure of elemental 1D and 2D materials. I recommend publishing the paper in Nature Communications in its present form.

We greatly appreciate the overwhelmingly positive response to our manuscript from the three reviewers, and thank them, for their time and efforts.

Reviewer 1 response:

We added yellow arrows to Supp. Fig. 1 indicating the 4 visible $(\pm 1, 0)$ and $(0, \pm 1)$ integer order spots and blue arrows indicating fractional order spots along the +X and +Y directions.